# Quasispecies Composition of Small Ruminant Lentiviruses Found in Blood Leukocytes and Milk Epithelial Cells

**DOI:** 10.3390/v13122497

**Published:** 2021-12-13

**Authors:** Monika Olech, Arkadiusz Bomba, Jacek Kuźmak

**Affiliations:** 1Department of Biochemistry, National Veterinary Research Institute, 24-100 Pulawy, Poland; jkuzmak@piwet.pulawy.pl; 2Department of Omics Analyses, National Veterinary Research Institute, 24-100 Pulawy, Poland; arkadiusz.bomba@piwet.pulawy.pl

**Keywords:** small ruminant lentiviruses (SRLVs), milk epithelial cells (MEC), single nucleotide variations (SNVs), next-generation sequencing (NGS)

## Abstract

Small ruminant lentiviruses (SRLVs) exist as populations of closely related genetic variants, known as quasispecies, within an individual host. The privileged way of SRLVs transmission in goats is through the ingestion of colostrum and milk of infected does. Thus, characterization of SRLV variants transmitted through the milk, including milk epithelial cells (MEC), may provide useful information about the transmission and evolution of SRLVs. Therefore, the aim of this study was to detect SRLVs in peripheral blood leukocytes (PBLs) and milk epithelial cells of goats naturally infected with SRLVs and perform single nucleotide variations analysis to characterize the extent of genetic heterogeneity of detected SRLVs through comparison of their *gag* gene sequences. Blood and milk samples from 24 seropositive goats were tested in this study. The double immunolabeling against p28 and cytokeratin demonstrated that milk epithelial cells originated from naturally infected goats were infected by SRLVs. Moreover, PCR confirmed the presence of the integrated SRLVs proviral genome indicating that MECs may have a role as a reservoir of SRLVs and can transmit the virus through milk. The blood and MEC derived sequences from 7 goats were successfully sequenced using NGS and revealed that these sequences were genetically similar. The MEC and blood-derived sequences contained from 3 to 30 (mean, 10.8) and from 1 to 10 (mean, 5.4) unique SNVs, respectively. In five out of seven goats, SNVs occurred more frequent in MEC derived sequences. Non-synonymous SNVs were found in both, PBLs and MEC-derived sequences of analyzed goats and their total number differed between animals. The results of this study add to our understanding of SRLVs genomic variability. Our data provides evidence for the existence of SRLVs quasispecies and to our knowledge, this is the first study that showed quasispecies composition and minority variants of SRLVs present milk epithelial cells.

## 1. Introduction

Small ruminant lentiviruses (SRLVs), comprising maedi-visna virus (MVV) and caprine arthritis encephalitis virus (CAEV) are RNA viruses belonging to the Lentivirus genus of the family Retroviridae which infect sheep and goats. SRLVs induce a persistent infection-causing progressive inflammatory lesions exerting arthritis, mastitis, chronic interstitial pneumonia, encephalitis, and progressive weakness [1]. The route of transmission is related to the body fluids. Lactogenic transmission via ingestion of colostrum and milk is considered to be the most important way of infection for SRLVs in kids while horizontal transmission via respiratory secretions play important role in older animals [1,2,3].

The provirus of SRLVs contains three structural genes (*gag*, *pol*, and *env*) and accessory genes (*vif, rev,* and *vpr*) flanked by non-coding long terminal repeat (LTR) sequences. The *pol* and *gag* genes are relatively conserved while the *env* gene shows a high level of variability. SRLVs are characterized by extensive genetic diversity. Based on classification proposed by Shah et al., [4] using the *gag-pol* (1.8 kb) and the *pol* (1.2 kb) region, SRLVs are divided into five major groups, A-E, which differ from each other in 25–37% of their nucleotide sequences. Groups A, B and E are further divided into several subtypes. However, due to low proviral load and high genetic variability of SRLV strains, more strains were added to the phylogeny based on shorter genomic fragments than those proposed by Shah et al. This leads to a current SRLVs phylogeny containing five groups and 29 subtypes.

Moreover, within infected individuals, SRLVs circulate not as a single genotype but as a cloud of closely related sequence variants called quasispecies. Analysis of variants isolated from different body compartments showed that the quasispecies are not randomly distributed but can be compartmentalized within different organs or tissues of the host [5,6,7,8]. Viral genetic heterogeneity, generated by single nucleotide variations (SNVs), is believed to be a strategy of virus evolution and adaptability. Moreover, the quasispecies nature of SRLVs may contribute to their transmission, especially favoring cross-species infection, and is critical for viral tissue tropism and pathogenesis [9].

Monocyte-macrophage lineage and dendritic cells are the main target cells for SRLVs. However, the tropism of the SRLVs for cells of the epithelial lineage has been widely demonstrated in a large variety of animal tissues including mammary glands and milk epithelial cells (MEC) [10,11,12,13,14]. Thus, not only macrophages but also epithelial cells may also act as a source of infection and play important role in virus dissemination. Additionally, these cells may contain a subpopulation of locally replicating variants that may be distinct from viruses detected in the peripheral blood and may act as founders for the descent generations. Thus, characterization of SRLV variants transmitted through milk epithelial cells, may provide useful information about the transmission and evolution of SRLVs.

In a previous study, we investigated the existence of differences in the population of SRLVs proviral DNA present in the peripheral blood leukocytes (PBLs) and colostrum cells of goats naturally infected with SRLV subtype A17. We found statistical evidence of compartmentalization between blood and colostrum although phylogenetic evidence of such compartmentalization was not obvious [8]. Although clone-based sequencing is more common method to study the quasispecies, it is labor-intensive and have several disadvantages, including the generation of limited information. This study extends our previous investigative approach, including next-generation sequencing (NGS), applied to perform single nucleotide variations analysis of *gag* gene sequences derived from PBLs and milk epithelial cells (MEC) of goats naturally infected with SRLVs representing different subtypes. By increasing the depth of sequencing, NGS enables the sequencing of genetic variants that are present at low frequencies in the virus population. Therefore, NGS data should better reflect the quasispecies composition of SRLVs population.

## 2. Materials and Methods

### 2.1. Animals and Sampling

Milk and blood samples were taken from serologically 24 positive goats originated from 3 flocks. The serological status of SRLVs infection was determined using the ELISA test (ID Screen MVV/CAEV Indirect Screening test, IDVet, Grabels, France) according to the manufacture’s protocol. Blood was taken by jugular venipuncture and collected in EDTA and serum tubes for further analysis. Samples of fresh goat milk were collected for the isolation of milk epithelial cells. Peripheral blood leukocytes (PBLs) isolation was performed as previously described [15]. All methods were performed in accordance with the relevant guidelines and regulations. Specifically, blood collection was approved (no 37/2016) by the Local Ethical Committee on Animal Testing at the University of Life Sciences in Lublin (Poland).

### 2.2. Milk Epithelial Cell (MEC) Isolation and Culture

Cells were isolated from 100 milliliters of milk from each goat by centrifugation at 500× *g* for 10 min at 4 °C. After washing cell pellets were resuspended in 10 mL RPMI medium with 10% FBS and 1% Antibiotic–Antimycotic (Thermo Fisher Scientific, Carlsbad, CA, USA), seeded into 25 cm^2^ tissue culture flasks, and incubated at 37 °C overnight. Next, non-adherent cells were washed out, a fresh culture medium was added, and cells were incubated at 37 °C, 5% CO_2_. After 2–3 weeks of culture, foci of milk epithelial cells emerged and, after three serial passages, homogenous MEC cultures were obtained [16].

### 2.3. Immunofluorescence

MEC were seeded in 4 well plates and after 24 h were fixed with cold acetone. After incubation for 20 min in 1% Triton X-100, diluted in PBS, cells were rinsed in 0.1% PBS-Tween 20 (PBST) and stabilized for 20 min with 0.1 M glycine solution. Double labeling against p28 and cytokeratin as a marker of epithelial cells in combination with nuclei staining was performed. The first viral protein was tagged by incubation for 10 h at 4 °C with monoclonal antibodies directed against CAEV Gag-p28 (CAEP5A1, VMRD, Pullman, Washington, DC, USA) diluted 1:1000 in PBS supplemented with 10% horse serum (Horse Serum Donor Herd, Sigma-Aldrich, Schnelldorf, Germany). After three washes with PBST, cells were incubated with the secondary antibody, 1:250 Alexa Fluor 555 donkey anti-mouse IgG (anti-mouse IgG H+L, Invitrogen, Eugene, OR, USA) for 2h at room temperature (RT). After washing, samples were fixed with 4% paraformaldehyde (PFA) for 15 min at RT, washed in PBST, and incubated with diluted 1:10 monoclonal antibodies directed against cytokeratin (Monoclonal Anti-Cytokeratin, pan (mixture) from the clones C-11, PCK-26, CY-90, KS-1A3, M20, A53-B/A2, Sigma-Aldrich, St Louis, MO, USA) overnight at 4 °C. After washing with PBST cells were incubated with the secondary antibody, Alexa Fluor 488 donkey anti-mouse IgG (anti-mouse IgG H+L, Invitrogen, Eugene, OR, USA) for 2 h at RT. Cells were then washed three times with PBST and counterstained with Hoechst 33,342 (Invitrogen, Eugene, OR, USA) before mounting with DAKO fluorescent medium (DAKO-S302380, Agilent, Santa Clara, CA, USA). Each sample was tested for the presence of non-specific labeling (negative control) by omitting the primary antibody incubation during the labeling experiment.

### 2.4. DNA Extraction

Total DNA was extracted from pelleted blood cells and MEC culture using the DNeasy Blood and Tissue Kit (Qiagen, Hilden, Germany) following the manufacturer’s instructions. DNA concentration and the 260 nm/280 nm ratio were measured using GeneQuant (GE Healtcare, Warsaw, Poland) and kept frozen at −20 °C until use.

### 2.5. PCR Amplification and Sanger Sequencing

DNA extracted from peripheral blood leukocytes (PBLs) and cultivated MEC were used for amplification of 625 bp fragment of the *gag* gene encoded capsid protein (CA) [15]. PCR products were detected by separation in 1.5% agarose gel and after purification were sequenced in both directions on the automated sequencer 3730xl DNA Analyzer (Applied Biosystems, Waltham, MA, USA) using Big Dye Terminator v3.1 Cycle Sequencing kit.

### 2.6. Next-Generation Sequencing (NGS) and Analysis of NGS Data

PCR products were also subjected to NGS sequencing. The quality and quantity of obtained PCR products were checked using Nanodrop 2000 Spectrophotometer (Thermo Scientific, Wilmington, DE, USA) and Quant-iT PicoGreen dsDNA Assay Kit (Thermo Fisher, Eugene, OR, USA). Then amplified products (30 ng) were fragmented using Covaris E220 (Covaris, Woburn, MA, USA) to an average fragment size of approximately 400pb. Barcode-tagged sequencing libraries were generated from the fragmented DNAs using the NEBNext Ultra DNA Library Prep Kit (New England Biolabs, Ipswich, MA, USA). The resulting individual DNA libraries were quality checked and quantified on the Agilent 2100 Bioanalyzer (Agilent Technologies, Santa Clara, CA, USA) using the High Sensitivity DNA kit (Agilent Technologies, Waldbronn, Germany) and quantified to greater accuracy using qPCR. Libraries were pooled and sequenced using the Illumina MiSeq platform with 2 × 300 bp paired-end reads (PE300). Primary data analysis was performed using MiSeq Reporter v2.5.1 (Illumina, San Diego, CA, USA) to generate a pair of FASTQ files for each sample. The FASTP [17] and Cutadapt software [18] were used to remove adapters, PCR primers and reads shorter than 30 base pairs and those with a Phred quality scores lower than 15. To identify SNVs within viral populations present in each sample, obtained trimmed reads were mapped to the appropriate sequences obtained by Sanger sequencing using BWA MEM [19] on default parameters. Alignment has been done and every sequence had the same length. FreeBayes v. 1.3.1 [20] and BCFtools [21] package were used to call and filter variants. The frequency distribution of SNVs was visualized using R 3.6.3 and ggplot2 [22]. We included only variants present in at least 5% of the reads at a given position. SNVs effects were predict by variant annotation package [23]. SRLVs mutation frequency was calculated by dividing the number of mutations (nucleotides/amino acids that differed from the consensus sequences) by the total number of nucleotides sequenced. NGS and Sanger sequencing was performed by a commercial company (Genomed S.A., Warsaw, Poland).

### 2.7. Phylogenetic Analysis

The obtained consensus sequences were aligned using CLUSTAL W and was then used to generate a phylogenetic tree using the maximum likelihood (ML) method. Nonparametric Bootstrap analysis with 1000 iterations was used to evaluate the robustness of evolutionary relationships. Alignment, model testing, and ML tree building were performed using the MEGA 6 application [24]. All novel sequences reported in this study were submitted to the Gen-Bank database under accession numbers: OK149652-OK149665. In order to analyze the evolutionary relationship, consensus sequences obtained in this study were aligned with other published sequences from GenBank representing the genotypes described to data (A–C and E). We included only sequences of length matching data obtained in this study.

## 3. Results

### 3.1. MEC Cultures and Fluorescence Results

Milk epithelial cells (MEC) were successfully isolated and cultivated from milk samples of 17 out of 24 goats (Table 1). The majority of cells had attached to the surface of the flasks during the initial 24h of culture and then formed discrete disseminated foci of epithelial cells. The initial monolayers were strewn with sparsely scattered macrophages which disappeared after the second to third passage. No cytopathic effect was observed in cell cultures. The epithelial nature of the cells was confirmed by confocal fluorescence microscopy. The data obtained with immunofluorescence using antibodies specific to the constitutively expressed cytokeratin markers showed that all milk epithelial cell cultures were cytokeratin marker positive (Figure 1A). Specific labeling with a monoclonal antibody specific for SRLV p28 Gag protein was also observed in all tested MEC cultures (Figure 1B). The double p28/cytokeratin immunolabeling revealed that the virus antigens were localized in the cytoplasm of epithelial cells (Figure 1C). Negative controls confirmed that fluorescence results were specific (Figure 1D).

### 3.2. Genetic Characterization of SRLV in PBLs and MEC

Total DNA isolated from peripheral blood leukocytes (PBLs) and MEC cultures originated from 17 goats were used as a template for PCR amplification of SRLV-*gag* region. The band corresponding to the expected 625 bp PCR product was detected in DNA from all 17 PBLs samples and in 12 (70.6%) samples, derived from MEC. In the case of five goats (#9509, #5871, #7660, #6021, and #5819) samples derived from PBLs were PCR positive while corresponding samples derived from MEC culture were negative (Table 1). Thus, these samples were not included in further analysis. Sanger sequencing of all PCR products was performed and BLAST searches at NCBI confirmed the presence of Small ruminant lentiviruses sequences. The phylogenetic tree (Figure 2) showed that obtained sequences clustered into four different subtypes, A17, A5, A12 and B2. In particular, PBLs and MEC-derived sequences from goats #1561, #5654 and #9431 originated from flock 1 were closely related to the sequences of variant #8344 (95.5–99.4%) representing subtype A17 while sequences from goats #5994, 5826, #5918, #5888, #5962, #4742, and #3038 from flock 3 were closely related with sequences of variants #6038 and #7592 representing subtype A5 (99.0–99.9%). PBLs and MEC-derived sequences from goat #8699 from flock 2 were closely related to the sequences of A12 Polish strain #5 (89.7–90.0%) while PBLs and MEC-derived sequences from goat #3540 from flock 2 were closely related to the sequences of strains #4084 and #4106 (97.5–97.6%) representing subtype B2. PBLs and MEC-derived sequences of each goat were closely related to each other and shared 99.0–100% nucleotide sequence identity.

Next, PBLs and MEC derived *gag* gene amplicons from 12 goats were subjected to the NGS sequencing but finally samples from 7 goats (#1561, #5918, #5888, #5962, #4742, #3038, and #8699) passed the stringent quality control (QC) and were successfully sequenced.

Obtained sequences were used to identify single nucleotide variations (SNVs) in order to determine the variability of SRLV sequences derived from PBLs and MEC. Multiple variation sites were identified in PBLs and MEC-derived sequences, with the highest number (*n* = 30) detected in MEC-derived sequences of goat #1561 and the lowest (*n* = 1) detected in PBLs-derived sequences of goat #5888 (Table 2, Figure 3).

The MEC and PBLs-derived sequences contained from 3 to 30 (mean, 10.8) and from 1 to 10 (mean, 5.4) unique SNVs, respectively. In five out of seven goats, SNVs occurred more frequently in MEC-derived sequences. In goat #4742 SNVs frequency in MEC and PBLs-derived sequences were at the same level while in goat #8699 the higher number of SNVs was detected in sequences derived from PBLs (Table 2). When sequences from all goats were compared, 50 SNVs were found only in MEC-derived sequences, 54 SNV positions were shared, and 17 were found only in PBLs-derived sequences. As shown in Figure 3, eight hot spot mutations, with more than four mutations in the same position, were identified. Non-synonymous SNVs were found in both, PBL and MEC-derived sequences of analyzed goats and their total number differed between animals. The highest number (*n* = 7) of non-synonymous SNVs, which resulted in a change in the amino acid, was detected in PBLs-derived sequences of goat #5918 and the lowest (*n* = 0) detected in blood-derived sequences of goat #1561 (Table 2, Figure 4). The number of non-synonymous SNVs ranged from 1 to 6 and from 0 to 7 in MEC and PBLs-derived sequences, respectively. There was no non-synonymous mutation in the sequences of immunodominant epitope 3 and only one non-synonymous SNV in PBLs-derived sequences of goat #5918 in epitope 2. In contrast, in the major homology region (MHR), non-synonymous SNVs were observed in four goats (#1561, #5918, #4742, and #3038) in PBLs and milk epithelial cell-derived sequences. The distribution of synonymous and non-synonymous SNVs are presented in Figure 4.

## 4. Discussion

Small ruminant lentiviruses, similar to other retroviruses, have been shown to exhibit high mutation rates primarily due to the low-fidelity of RNA polymerase. Therefore, these viruses exist as populations of closely related genetic variants, known as quasispecies, within an individual host. Since the privileged way of SRLVs transmission in goats is through the ingestion of colostrum and milk of infected ewes, special attention should be paid to the role and significance of milk epithelial cells in SRLVs transmission and their heterogeneity.

Several in vitro studies have shown that epithelial cells, including cells from the mammary gland [12,13], genital track [26,27,28], and granulosa cells [29] are highly susceptible to productive infection with SRLVs. SRLVs proviral DNA has also been detected in bronchial epithelial cells [30] and epithelial cells of the intestinal crypts [31]. However, the contribution of milk epithelial cells in SRLVs replication and transmission remains largely unknown. The double immunolabeling against p28 and cytokeratin, performed in this study, clearly demonstrated that milk epithelial cells originated from naturally infected goats were infected by SRLVs. Moreover, PCR confirmed the presence of the integrated SRLVs proviral genome. This clearly indicates that MECs may have a role as a reservoir of SRLVs and can transmit the virus through milk. Lack of cytopathic effect observed in MEC cultures in this study is in accordance with observations of other authors [14,16], who showed constitutive virus production in naturally infected milk epithelial cells in absence of any cytopathic effect. It is unclear whether the infection of milk epithelial cells occurred directly by virus particles or indirectly through cell-to-cell contact with infected macrophages, which are the main target of SRLVs in vivo. The presence of budding particles at the MEC cell membrane suggests that MECs may give rise to infective virus particles [32]. However, it is more likely that MECs would be rather infected by random fusion with SRLV-infected macrophages, passing through the tissue [28]. Because, as we show in this study, the primary cultures contained cells of the monocyte/macrophages lineage, and mature macrophages were seen on the culture at two passages, we can’t exclude the hypothesis that viral infection of MEC occurred *in vitro*. Our results demonstrated that in vitro cultures of MEC were sensitive to infection with a natural field strain of SRLV but not determined the in vivo source of infection

It is also possible that vesicular epithelial particles, carrying viral proteins or nucleic acids, may induce the spread of infection by improving and sustaining the production of a virus in infected cells, as was shown for human immunodeficiency virus (HIV) and Epstein–Barr virus (EBV) infections [33]. It is presumed that SRLVs may enter the MECs but the infectious virus may be produced after cell activation by host factors [11]. Thus, it is possible that infected milk epithelial cells may constitute a latent SRLVs reservoir which are involved in virus reactivation during latency.

Restrictions on the ability of the virus to migrate freely between different cells, tissues, and organs can have a profound effect on viral diversity. A study on SRLVs compartmentalization performed to date revealed the presence of different viral sequences in the blood, mammary gland, lungs, and central nervous system tissue in clinically affected sheep [6] and in the colostrum and peripheral blood of goats [5,8]. Similarly, compartmentalization has been well documented in the brain, cerebrospinal fluid, and genital tract, as well as in the gut, lung, liver, kidney, and breast milk of patients infected with HIV-1 [34,35,36,37,38]. In the current study, SRLVs sequences from PBLs and milk epithelial cells were genetically similar suggesting an equilibrium of virus between these compartments. However, in our study, we not analyzed highly variable regions V1V2 or V4V5 of *env* gene but a fragment of the *gag* gene encoding capsid protein which is highly conserved and differences in this region would be less likely due to high sequence homology. Although, the numbers of SNVs detected in MEC and PBLs-derived sequences differed between animals, in five out seven goats, the higher numbers of SNVs were identified in sequences derived from MEC. It was a little bit of a surprise because consistent with migration of the virus from blood to other tissues and vice versa it is thought that PBLs-derived sequences should display the broadest variation. So, it is hypothesized that higher mutation rates in MEC-derived samples allowing the appearance of new quasispecies with potentially beneficial mutations, which afford a greater probability to evolve and adapt to a new host. There is strong evidence to support that transmission results in a population bottleneck with only a small fraction of the variants passing to the new host [7]. Pisoni et al., revealed that subtype A10 was more efficiently transmitted from co-infected goats to the offspring than goat-specific subtype B1. In our study, goats were infected by subtypes A5, A12, and A17, which are closely related to the strain K1514 originally identified as sheep-specific strain. This may indicate that viruses that cross the species barrier may have a higher “fitness”, facilitating effective transmission from the dam to kids However, more studies including analysis of other fragments of the SRLVs genome are required to support this hypothesis. On the other hand, our results indicated that the number of SNVs differed between animals what may suggest that other factors may also influence the variability of SRLVs. Our results revealed that in 92.8% of analyzed sequences all available guanine bases were substituted by adenines suggesting that host factors such as APOBEC3 and immune responses exerting a selective pressure may contribute to the generation and evolution of the quasispecies [9,39]. Because the samples analyzed in this study were taken at a single time point from early and late-lactating goats we also cannot rule out the possibility that SRLVs diversity changes significantly over time. It can be also assumed that degree of variation of SNVs in MEC-derived samples has been a little changed after three passages in culture, but future analysis is essential to confirm this hypothesis.

A number of synonymous and nonsynonymous SNVs obtained on the basis of PBL and MEC-derived sequences varied between animals. Synonymous mutations were the most frequent type of point mutation. Only in one goat (#5962) had a number of non-synonymous SNVs higher than the number of synonymous mutations, in both MEC and PBLs-derived sequences, in what may be interpreted as strong evidence of positive selection. Moreover, comparing sequences derived from PBLs and MEC of all goats, eight hot spot mutations were identified. Such mutations that recur at the same genomic site, may also be interpreted as a result of selection pressure exerted by the host immune systemIn our study, we did not observe variation in the sequence of epitope 2 and 3 which confirmed that sequences of these epitopes are extremely conserved. Only in the major homology region (MHR), single non-synonymous SNVs were observed in the PBLs and milk epithelial cell-derived sequence of four goats. The direct influence of these SNVs is unknown but mutation in this region may impair assembly of the viral core and change virus infectivity [40,41].

In conclusion, the present study confirmed that milk epithelial cells can be targets of SRLVs and sources of SRLVs proviral DNA and support the possibility that SRLVs could be spread by milk epithelial cells. The results of this study add to our understanding of SRLVs genomic variability. Our data provides evidence for the existence of SRLVs quasispecies and to our knowledge, this is the first study that showed quasispecies composition and minority variants of SRLVs present milk epithelial cells.

## Figures and Tables

**Figure 1 viruses-13-02497-f001:**
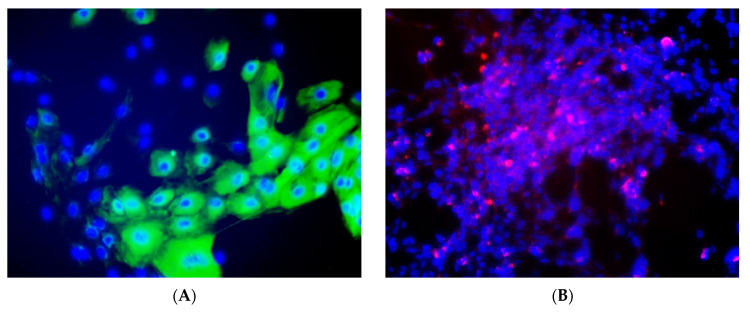
Immunostaining of CAEV antigen and the cytokeratin epithelial marker in milk epithelial cells originated from goats examined in this study. (**A**) single cytokeratin labeling (green color) (**B**) single CAEV p28 labeling (red color) (**C**) double CAEV p28 (red color) and cytokeratin (green color) labeling on the same samples (**D**) cells stained by omitting the primary antibody incubation during labeling.

**Figure 2 viruses-13-02497-f002:**
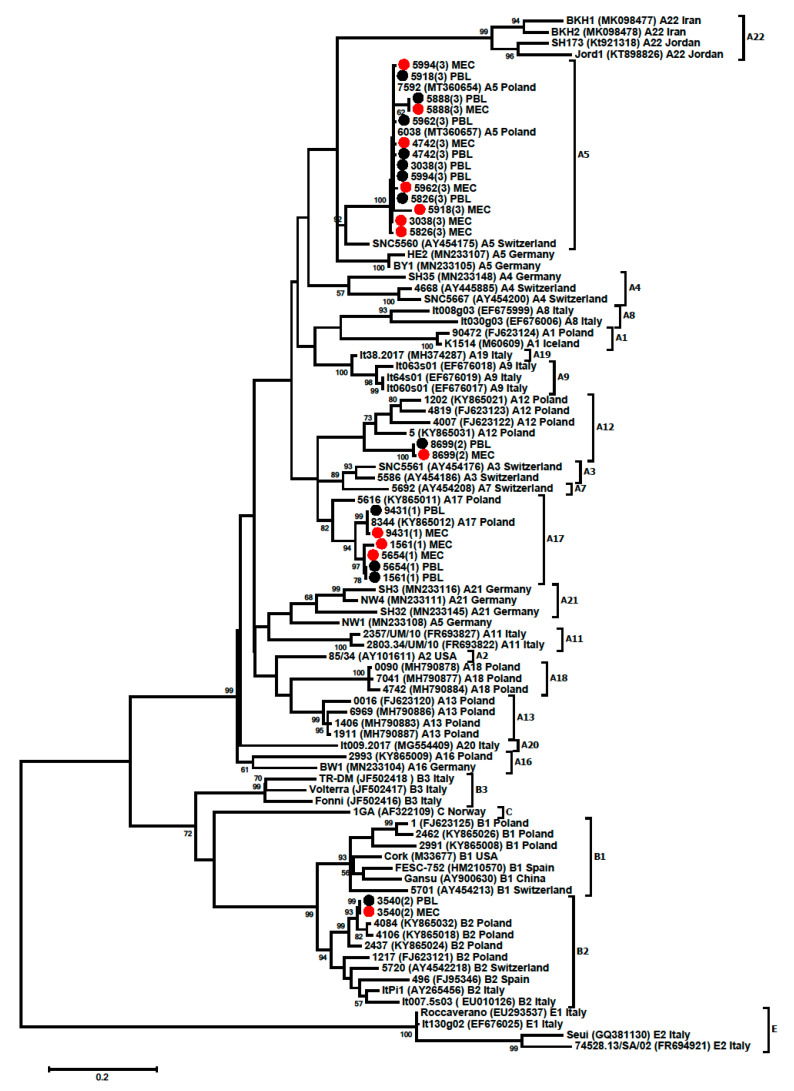
Maximum likelihood phylogenetic tree based on *gag* sequences of SRLV isolates. In the present study, the SRLV subtypes found by Colitti et al. [25] were renamed from A18 to A19 and from A19 to A20. SRLVs sequences isolated from goats examined in this study are indicated with filled circles: red color-milk epithelial cells (MEC), black color-peripheral blood leukocytes (PBLs) followed by the flock origin (1–3). The reference sequences are indicated by name and their GenBank accession number between brackets followed by the subtype and country origin.

**Figure 3 viruses-13-02497-f003:**
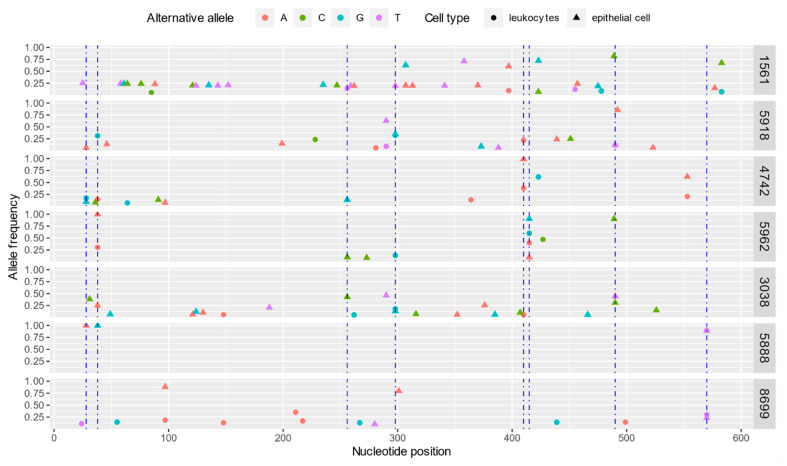
Distribution of single nucleotide variants (SNVs) in SRLVs gag genesequences encoding capsid protein derived from PBLs and milk epithelial cells (MEC) of goats analyzed in this study. The blue dotted lines indicate the hot spot mutations, where the number of SNVs was ≥4.

**Figure 4 viruses-13-02497-f004:**
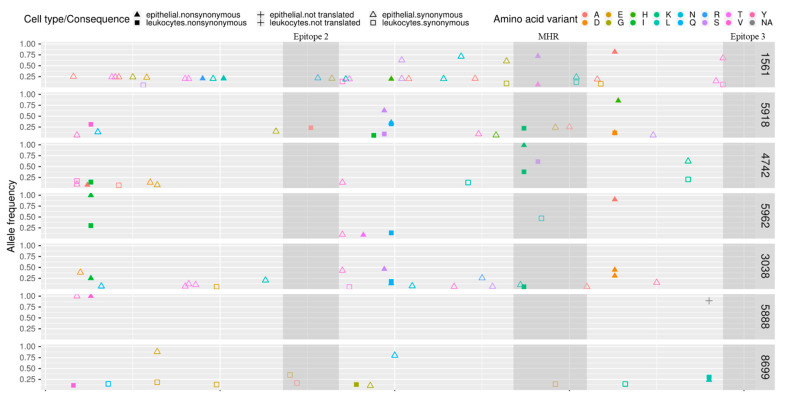
Distribution of synonymous and non-synonymous single nucleotide variants (SNVs) in SRLVs *gag* gene sequences encoding capsid protein derived from PBLs and milk epithelial cells of goats analyzed in this study. MHR-major homology region.

**Table 1 viruses-13-02497-t001:** Characteristic of samples examined in this study.

Flock	Goat	MEC Culture	PCR MEC	Presence of p28 in MEC	PCR PBL	SRLVs Subtype
1	9431	+	+	+	+	A17
	5654	+	+	+	+	A17
	1561 *	+	+	+	+	A17
2	8699 *	+	+	+	+	A12
	3540	+	+	+	+	B2
	9509	+	−	nt	+	nt
	0580	−	nt	nt	+	nt
	0788	−	nt	nt	+	nt
	9510	−	nt	nt	+	nt
	3533	−	nt	nt	+	nt
	3535	−	nt	nt	+	nt
	0599	−	nt	nt	+	nt
3	4742 *	+	+	+	+	A5
	5994	+	+	+	+	A5
	5871	+	−	nt	+	nt
	5826	+	+	+	+	A5
	7592	+	−	nt	+	nt
	5962 *	+	+	+	+	A5
	7660	−	nt	nt	+	nt
	6021	+	−	nt	+	nt
	3038 *	+	+	+	+	A5
	5888 *	+	+	+	+	A5
	5918 *	+	+	+	+	A5
	5819	+	−	nt	+	nt

MEC—milk epithelial cells; PBL—peripheral blood leukocytes; nt—not tested; *—samples used for NGS.

**Table 2 viruses-13-02497-t002:** Characteristic of SRLVs sequences derived from PBLs and milk epithelial cells (MEC) of goats analyzed in this study.

Goat	SRLVs Subtype	Origin	Number of SNVs	SNVs Frequency	Syn	N-Syn	N-Syn Frequency
1561	A17	MEC	30	5.20%	24	6	3.14%
		PBLs	6	1.05%	6	0	0.00%
8699	A12	MEC	4	0.70%	3	1	0.52%
		PBLs	10	1.74%	7	3	1.57%
4742	A5	MEC	7	1.22%	5	2	1.05%
		PBLs	7	1.22%	4	3	1.57%
5962	A5	MEC	4	0.87%	1	3	1.57%
		PBLs	3	0.70%	1	2	1.05%
5918	A5	MEC	12	2.09%	8	4	2.09%
		PBLs	7	1.22%	0	7	3.66%
5888	A5	MEC	3	0.52%	2	1	0.52%
		PBLs	1	0.17%	Not translated *		-
3038	A5	MEC	19	3.31%	14	5	2.62%
		PBLs	4	0.87%	2	2	1.05%

MEC—milk epithelial cells; PBL—peripheral blood leukocytes; SNVs—single nucleotide variants; Syn—synonymous; N—Syn-non-synonymous; * translating code containing degenerate nucleotides.

## Data Availability

All data generated and analyzed in this study are included in this article.

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
