# Peer review of "Quasispecies Composition of Small Ruminant Lentiviruses Found in Blood Leukocytes and Milk Epithelial Cells"

_viruses, 2021, doi:10.3390/v13122497_

Round 1
Reviewer 1 Report
In this article, the authors characterize the SRLV variants in blood leukocytes (PBLs) and milk epithelial cells (MEC) from 24 seropositive goats. The presence of p28 and provirus (by PCR) is detected in cultured MEC. The PBL and MEC gag sequences of 7 goats are sequenced by NGS, and the results show that they are genetically similar. A different number of non-synonymous single nucleotide variations are found in the PBL and MEC sequences. All the results together suggest a role for MEC in SRLV transmission and the existence of quasispecies and minority variants in MEC, which had not been previously described.
The main contribution of the work is the detection of p28 and provirus in cultured ECM and the evidence of gag variants in PBL and ECM sequences, although it is a highly conserved gene in SRLV. The implications of these results are not known. The objective of this manuscript is not really new, as there are several publications describing SRLV compartmentalization in infected sheep and goats with results very similar, including one from the same authors analysing the gag sequences from PBLs and colostrum somatic cells (Olech and Kuźmak, 2019)
The title is not very descriptive as genetic heterogeneity of SRLVs is to be expected in this type of virus. Why a note?
Throughout the document, use PBL and milk epithelial cells (MEC) instead of blood and epithelial cells; also review the use of SNV and / or SNP
Introduction
The authors should better explain how phylogenetic analysis are performed to establish SRLV groups and subtypes; and include more references on previous studies of viral compartmentalization and variants and quasispecies of env and gag sequences, especially comparing colostrum and PBL.
The authors must modify the description of a provirus structure to: “The provirus contains three structural genes and accessory genes flanked by non-coding long terminal repeat (LTR) sequences “.
The authors have previously described the SRLV compartmentalization between blood and colostrum (env, gag, and LTR sequences) although phylogenetic evidence of such compartmentalization was not obvious (Olech and Kuźmak, 2019), so they should better highlight what is new compared to their previous results and compare with results those obtained in the current study on ECM. Furthermore, they should more clearly justify why this study focuses only on the gag sequences of PBLs and MECs.
Material and methods
Do the animals have any clinical signs, especially mastitis, associated with the infection? It could have some relationship with the different sequences observed in PBL and MEC.
Culture of MEC: Is an antibiotic used in the culture medium of cells?
Results and discussion
Table 1: Change column title from "presence of p28" to "presence of p28 in MEC". In eight samples in which no ECM culture has been obtained (0580, 0788, 9510,3533, 3535,0599,7660, 5819), it is impossible to perform PCR on these cells. Therefore, the PCR result should be “nt” rather than negative.
It is difficult to understand the information of the subtypes identified in each flock. Authors can add this information in Table 1 in a new column and write it clearly in the text.
Figure 2: The sequences included in each subtype can be circled and the subtype they resemble written next to them. That would facilitate the interpretation of the figure. Please, write gag on italics in figure 2 legend. Identify the sequences of PBL and MEC with different colored circles instead of black circles, thus the results obtained from PBL and MEC can be better interpreted and compared. If possible, Figure 2 would be more complete if the flock is added and, therefore, it can be related to the identified subtypes. What is the meaning of the number in parentheses next to the SRLV isolate number?
Lane 232: Authors should specify that the NGS results have been obtained from the gag genes
Lanes 252-270: It does not appear that the authors have found a pattern in the viral sequences of PBLs and MEC although the higher numbers of SNVs (and/or SNPs) seem to be identified in sequences derived from MEC which could be related to new quasispecies (lanes 330-333); the results seems to depend on each goat analyzed.
Table 2: In the tittle, please correct “from blood and epithelial cells of goats” to “from PBLs and milk epithelial cells (MEC) of goats. Also, in the origin column, change epithelial cell to MEC and leukoytes to PBLs (as in the rest of the document). Add the meaning of PBL and MEC to the legend of the table. The authors write SNVs and SNPs, but it is unclear if they mean the same thing. It is better to call them SNVs, as it is written in the legend of the table.
Figures 3 and 4 are difficult to interpret (the symbols are very small, and the differences are not clear; the dotted line appears black instead of blue. The isolate numbers in the figure should be identified.). It should be added that the sequences are from the gag gene (and it would be interesting to know which region exactly). Again, in the figure legend correct “from blood and epithelial cells of goats” to “from PBLs and milk epithelial cells (MEC) of goats.
Lane 325-356: “In the current study, SRLVs sequences from blood and epithelial cells were genetically similar suggesting an equilibrium of virus between these compartments” This result seems to contradict the compartmentalization in PBLs and MEC proposed by the authors.
It should be better explained why it is so interesting to analyze the variations in the gag gene if it is so conserved in SRLVs and the implications it may have on the pathogenesis of these viruses.
It is interesting the conclusion proposed by the authors of a higher fitness in the subtypes more similar to those specific of sheep, although the data of the study are not enough to verify this. It would be also interesting to know the variations in env and LTR in the goats analyzed.
The last final paragraph does not seem to add much new data to what is already known about SRLVs.
Author Response
We would like thank the reviewer for his comments on our manuscript. We have acted upon the suggestions provided by the reviewer and alterations were included in the updated version of the manuscript.
Reviewer 1
In this article, the authors characterize the SRLV variants in blood leukocytes (PBLs) and milk epithelial cells (MEC) from 24 seropositive goats. The presence of p28 and provirus (by PCR) is detected in cultured MEC. The PBL and MEC gag sequences of 7 goats are sequenced by NGS, and the results show that they are genetically similar. A different number of non-synonymous single nucleotide variations are found in the PBL and MEC sequences. All the results together suggest a role for MEC in SRLV transmission and the existence of quasispecies and minority variants in MEC, which had not been previously described.
The main contribution of the work is the detection of p28 and provirus in cultured ECM and the evidence of gag variants in PBL and ECM sequences, although it is a highly conserved gene in SRLV. The implications of these results are not known. The objective of this manuscript is not really new, as there are several publications describing SRLV compartmentalization in infected sheep and goats with results very similar, including one from the same authors analysing the gag sequences from PBLs and colostrum somatic cells (Olech and Kuźmak, 2019)
The title is not very descriptive as genetic heterogeneity of SRLVs is to be expected in this type of virus. Why a note?
Re: The title has been changed.
Throughout the document, use PBL and milk epithelial cells (MEC) instead of blood and epithelial cells; also review the use of SNV and / or SNP
Re: It has been corrected.
Introduction
The authors should better explain how phylogenetic analysis are performed to establish SRLV groups and subtypes;
Re: The information has been included.
and include more references on previous studies of viral compartmentalization and variants and quasispecies of env and gag sequences, especially comparing colostrum and PBL.
Re: It has been corrected. There are only a few publications on SRLVs compartmentalization.
The authors must modify the description of a provirus structure to: “The provirus contains three structural genes and accessory genes flanked by non-coding long terminal repeat (LTR) sequences “.
Re: It has been corrected.
The authors have previously described the SRLV compartmentalization between blood and colostrum (env, gag, and LTR sequences) although phylogenetic evidence of such compartmentalization was not obvious (Olech and Kuźmak, 2019), so they should better highlight what is new compared to their previous results and compare with results those obtained in the current study on ECM. Furthermore, they should more clearly justify why this study focuses only on the gag sequences of PBLs and MECs.
Re: The information has been added to the introduction. The authors' original goal was to analyze the env fragment, but due to the low concentration of DNA derived from milk epithelial cell cultures we had problems with amplification of env fragment or the amount of PCR products was insufficient to analyse the samples by NGS.
Material and methods
Do the animals have any clinical signs, especially mastitis, associated with the infection? It could have some relationship with the different sequences observed in PBL and MEC.
Re: To the authors' knowledge, all goats used in this study did not show any clinical manifestation, including mastitis.
Culture of MEC: Is an antibiotic used in the culture medium of cells?
Re: Yes. This information has been added in section 2.2. Milk epithelial cell (MEC) isolation and culture
Results and discussion
Table 1: Change column title from "presence of p28" to "presence of p28 in MEC". In eight samples in which no ECM culture has been obtained (0580, 0788, 9510,3533, 3535,0599,7660, 5819), it is impossible to perform PCR on these cells. Therefore, the PCR result should be “nt” rather than negative.
Re: It has been corrected.
It is difficult to understand the information of the subtypes identified in each flock. Authors can add this information in Table 1 in a new column and write it clearly in the text.
Re: The information has been added in Table 1. Description in text is in lines 227-236.
Figure 2: The sequences included in each subtype can be circled and the subtype they resemble written next to them. That would facilitate the interpretation of the figure. Please, write gag on italics in figure 2 legend. Identify the sequences of PBL and MEC with different colored circles instead of black circles, thus the results obtained from PBL and MEC can be better interpreted and compared. If possible, Figure 2 would be more complete if the flock is added and, therefore, it can be related to the identified subtypes. What is the meaning of the number in parentheses next to the SRLV isolate number?
Re: It has been corrected. The reference sequences are indicated by name and their GenBank accession number between brackets.
Lane 232: Authors should specify that the NGS results have been obtained from the gag genes
Re: It has been added.
Lanes 252-270: It does not appear that the authors have found a pattern in the viral sequences of PBLs and MEC although the higher numbers of SNVs (and/or SNPs) seem to be identified in sequences derived from MEC which could be related to new quasispecies (lanes 330-333); the results seems to depend on each goat analyzed.
Re: The authors did not state that they found a pattern in the viral sequences of PBLs and MEC. We agree with the rewiever that the numer of SNVs seems to depend on each goat analyzed and such sentence has been added in discussion.
Table 2: In the tittle, please correct “from blood and epithelial cells of goats” to “from PBLs and milk epithelial cells (MEC) of goats. Also, in the origin column, change epithelial cell to MEC and leukoytes to PBLs (as in the rest of the document). Add the meaning of PBL and MEC to the legend of the table. The authors write SNVs and SNPs, but it is unclear if they mean the same thing. It is better to call them SNVs, as it is written in the legend of the table.
Re: It has been corrected.
Figures 3 and 4 are difficult to interpret (the symbols are very small, and the differences are not clear; the dotted line appears black instead of blue. The isolate numbers in the figure should be identified.). It should be added that the sequences are from the gag gene (and it would be interesting to know which region exactly). Again, in the figure legend correct “from blood and epithelial cells of goats” to “from PBLs and milk epithelial cells (MEC) of goats.
Re: It has been corrected.
Lane 325-356: “In the current study, SRLVs sequences from blood and epithelial cells were genetically similar suggesting an equilibrium of virus between these compartments” This result seems to contradict the compartmentalization in PBLs and MEC proposed by the authors.
Re: The authors assumed that the differences in SRLVs between PBLs and milk epithelial cells (MEC) would be greater. However, the results did not confirm this.
It should be better explained why it is so interesting to analyze the variations in the gag gene if it is so conserved in SRLVs and the implications it may have on the pathogenesis of these viruses.
Re: The explanation was mentioned above.
It is interesting the conclusion proposed by the authors of a higher fitness in the subtypes more similar to those specific of sheep, although the data of the study are not enough to verify this. It would be also interesting to know the variations in env and LTR in the goats analyzed.
Re: We agree with the rewiever. An appropriate annotation was included in the discussion.
Reviewer 2 Report
General considerations:
The submitted manuscript is well written and centered on a very relevant question of the SRLV biology, i.e., the importance of the mammary gland as a reservoir and source of infectious virus. The authors sequenced and phylogenetically analyzed a relevant number of provirus fragments derived from the PBMC and in vitro passaged milk epithelial cells obtained from naturally SRLV-infected goats. Additionally, they performed NGS on a subset of these proviruses and analyzed the detected SNPs within and between animals. The results are significant; however, their interpretation needs some clarifications.
Specific criticism:
The main criticism concerns the origin of the analyzed proviruses. The authors propose MEC as the source of these viruses. This is a crucial point, and the data shown don’t permit the authors to draw this conclusion (lines 296-298). An alternative scenario mentioned by the authors in their discussion (lines 306-309) is the transmission of the virus from so-called “accessory cells” to the in vitro passaged MEC. Indeed, this appears to be the potentially more plausible scenario. In fact, MEC isolated ex-vivo were passaged three times in vitro before analyzing the SRLV provirus sequences. During the first two passages, macrophages, the principal recognized source of infectious virus, were present in the cultures (lines 183-184) and disappeared only after the third passage. These macrophages likely infected the MEC. Puzzling is also the lack of cytopathic effect in these cells (lines 184-185) that in reference 9. was described as prominent in vitro infected MEC. How to solve this conundrum that is central to the message conveyed by this manuscript? In addition to the inquiries about the lack of cytopathic effect in MEC (lines 184-185), it would be important to know if the authors detected infectious virus in the supernatant of these cells. Did they titrate the virus in the supernatant and perform a quantitative RT-PCR looking for viral RNA?
The first experiment would be to look at the MEC immediately after isolation. Immunohistochemistry may not be sensitive enough at this point; however, a sensitive technique such as RNAScope or in situ hybridization would demonstrate the presence of SRLV RNA in MEC freshly isolated ex-vivo. In situ PCR-associated immunohistochemistry, as described in reference 10, should also detect ex-vivo infected MEC. Finally, single-cell RNA sequencing of cells freshly isolated from the mammary gland would prove or contradict the conclusions drawn by the authors.
If these techniques are not practicable, I would suggest rewriting the manuscript presenting the two possibilities, such as infection of MEC in vivo or in vitro by macrophages, as alternative scenarios that cannot be separated based on the data obtained.
Minor points:
Line 14: doe may be more appropriate than ewe.
Lines 67-70: “… under selective pressure” such as? Why should the viruses become more evolutionary stable?
Line 81: “positive 24 goats” should be “24 positive goats”
Lines 218 and 220: it may be more appropriate to use the term “variant” than “strain.”
Line 344: If APOBEC3 is involved, the authors should observe a dominant G to A pattern of mutations. Did that occur?
Line 360: Suggesting a neutralizing antibody pressure on Gag may be misleading.
Author Response
We would like thank the reviewer for his comments on our manuscript. We have acted upon the suggestions provided by the reviewer and alterations were included in the updated version of the manuscript.
Reviewer 2
General considerations:
The submitted manuscript is well written and centered on a very relevant question of the SRLV biology, i.e., the importance of the mammary gland as a reservoir and source of infectious virus. The authors sequenced and phylogenetically analyzed a relevant number of provirus fragments derived from the PBMC and in vitro passaged milk epithelial cells obtained from naturally SRLV-infected goats. Additionally, they performed NGS on a subset of these proviruses and analyzed the detected SNPs within and between animals. The results are significant; however, their interpretation needs some clarifications.
Specific criticism:
The main criticism concerns the origin of the analyzed proviruses. The authors propose MEC as the source of these viruses. This is a crucial point, and the data shown don’t permit the authors to draw this conclusion (lines 296-298). An alternative scenario mentioned by the authors in their discussion (lines 306-309) is the transmission of the virus from so-called “accessory cells” to the in vitro passaged MEC. Indeed, this appears to be the potentially more plausible scenario. In fact, MEC isolated ex-vivo were passaged three times in vitro before analyzing the SRLV provirus sequences. During the first two passages, macrophages, the principal recognized source of infectious virus, were present in the cultures (lines 183-184) and disappeared only after the third passage. These macrophages likely infected the MEC. Puzzling is also the lack of cytopathic effect in these cells (lines 184-185) that in reference 9. was described as prominent in vitro infected MEC. How to solve this conundrum that is central to the message conveyed by this manuscript? In addition to the inquiries about the lack of cytopathic effect in MEC (lines 184-185), it would be important to know if the authors detected infectious virus in the supernatant of these cells. Did they titrate the virus in the supernatant and perform a quantitative RT-PCR looking for viral RNA?
The first experiment would be to look at the MEC immediately after isolation. Immunohistochemistry may not be sensitive enough at this point; however, a sensitive technique such as RNAScope or in situ hybridization would demonstrate the presence of SRLV RNA in MEC freshly isolated ex-vivo. In situ PCR-associated immunohistochemistry, as described in reference 10, should also detect ex-vivo infected MEC. Finally, single-cell RNA sequencing of cells freshly isolated from the mammary gland would prove or contradict the conclusions drawn by the authors.
If these techniques are not practicable, I would suggest rewriting the manuscript presenting the two possibilities, such as infection of MEC in vivo or in vitro by macrophages, as alternative scenarios that cannot be separated based on the data obtained.
Re: The authors share the reviewer's comments relating to the question of the source of SRLV infecting MEC. Since it is not possible to perform additional experiments (lack of testing material), the authors made corrections in the discussion.
In reference 9 two variants of CAEV were used to infect mammary gland cells cultures originated from noninfected goats. In our study milk epithelial cells cultures were isolated from milk of naturally infected goats. Our results are in a line with results obtained by Mselli-Lakhal et al., 1999, where no cytopathic effect was observed in milk epithelial cells .Virus production was constitutive and occured in absence of any cytopathic effect in naturally infected cells and in those experimentally infected by one of two laboratory viruses.
Minor points:
Line 14: doe may be more appropriate than ewe.
Re: It has been corrected.
Lines 67-70: “… under selective pressure” such as? Why should the viruses become more evolutionary stable?
Re: The sentence has been rewrited.
Line 81: “positive 24 goats” should be “24 positive goats”
Re: It has been corrected.
Lines 218 and 220: it may be more appropriate to use the term “variant” than “strain.”
Re: It has been corrected.
Line 344: If APOBEC3 is involved, the authors should observe a dominant G to A pattern of mutations. Did that occur?
Re: Our results revealed that in 92.8% of analyzed sequences all available guanine bases were substituted by adenines suggesting that host factors like APOBEC3 and immune responses exerting a selective pressure may contribute to the generation and evolution of the quasispecies. This information was included in Discussion.
Line 360: Suggesting a neutralizing antibody pressure on Gag may be misleading.
Re: The sentence has been removed.
Round 2
Reviewer 2 Report
The authors convincingly answered the questions raised by the reviewers, and the modified interpretation of the data presented is sound.